# Genetic and Phenotypic Spectrum of KBG Syndrome: A Report of 13 New Chinese Cases and a Review of the Literature

**DOI:** 10.3390/jpm12030407

**Published:** 2022-03-05

**Authors:** Fenqi Gao, Xiu Zhao, Bingyan Cao, Xin Fan, Xiaoqiao Li, Lele Li, Shengbin Sui, Zhe Su, Chunxiu Gong

**Affiliations:** 1Department of Endocrinology, Genetics, Metabolism and Adolescent Medicine, Beijing Children’s Hospital, Capital Medical University, National Center for Children’s Health, Beijing 100045, China; drgfq77@126.com (F.G.); caoby1982@163.com (B.C.); lixq008@163.com (X.L.); lilele2006@163.com (L.L.); shengbinsui1998@163.com (S.S.); 2Department of Endocrinology, Shenzhen Children’s Hospital, Shenzhen 518000, China; zoeman221@163.com; 3Pediatric Dapartment, The Second Affiliated Hospital of Guangxi Medical University, Nanning 510000, China; fanxin602@163.com

**Keywords:** rare disease, KBG syndrome, *ANKRD11* gene, genotype–phenotype relationship, development retardation

## Abstract

KBG syndrome (KBGS) is a rare autosomal dominant inherited disease that involves multiple systems and is associated with variations in the ankyrin repeat domain 11 (*ANKRD11*) gene. We report the clinical and genetic data for 13 Chinese KBGS patients diagnosed by genetic testing and retrospectively analyse the genotypes and phenotypes of previously reported KBGS patients. The 13 patients in this study had heterozygous variations in the *ANKRD11* gene, including seven frameshift variations, three nonsense variations, and three missense variations. They carried 11 variation sites, of which eight were previously unreported. The clinical phenotype analysis of these 13 patients and 240 previously reported patients showed that the occurrence rates of craniofacial anomalies, dental anomalies, global developmental delays, intellectual disability/learning difficulties, limb anomalies, and behavioural anomalies were >70%. The occurrence rates of short stature, delayed bone age, and spinal vertebral body anomalies were >50%. The frequency of global developmental delays and intellectual disability/learning difficulties in patients with truncated *ANKRD11* gene variation was higher than that in patients with missense variation in the *ANKRD11* gene (*p* < 0.05). Collectively, this study reported the genotypic and phenotypic characteristics of the largest sample of KBGS patients from China and discovered eight new *ANKRD11* gene variations, which enriched the variation spectrum of the *ANKRD11* gene. Variation in the *ANKRD11* gene mainly caused craniofacial anomalies, growth and developmental anomalies, skeletal system anomalies, and nervous system anomalies. Truncated variation in the *ANKRD11* gene is more likely to lead to global growth retardation and intellectual disability/learning difficulties than missense variation in *ANKRD11*.

## 1. Introduction

KBG syndrome (KBGS, OMIM#148050) is a multisystem disorder, and “KBG” is derived from the initials of the surnames of three families who were first diagnosed with the disease in 1975 [1]. Since then, more than 300 cases have been reported. The typical manifestations are obvious craniofacial anomalies, macrodontia of the upper central incisors, skeletal anomalies, short stature, and growth and mental retardation. In addition, hearing loss, congenital heart disease (CHD), cryptorchidism, and behavioural anomalies are common (similar to Cornelia de Lange syndrome, CdLS). The syndrome showed complete dominance, with varying degrees of expressivity between and within families. In 2011, Sirmaci [2] confirmed ankyrin repeat domain 11 (*ANKRD11*) as the pathogenic gene responsible for the main phenotype of KBGS.

The ANKRD11 protein, containing 2663 amino acids, is widely expressed in the brain and is located mainly in neurons and glial nuclei. It acts as a key chromatin regulator that controls histone acetylation and gene expression during nerve development and plays an important role in nerve plasticity [2]. ANKRD11 contains multiple functional domains, including two transcriptional inhibition domains at the N- and C-terminals, one transcriptional activation domain, one ankyrin repeat domain, and multiple nuclear localization signals [3,4]. The N-terminal domain may be related to protein interactions and homodimer synthesis, while the C-terminal domain contains an important domain for ANKRD11 protein degradation. It is generally believed that KBGS is caused by truncated variation affecting the N-terminus of ANKRD11, triggering nonsense-mediated mRNA decay and leading to haploinsufficiency of *ANKRD11* [2]. However, in patients who carry a variant that leaves the N-terminus of the protein intact, a dominant-negative effect on the cell cycle is the main pathogenic mechanism of KBGS [5]. ANKRD11 inhibits ligand-dependent transcriptional activation by recruiting histone deacetylase to interact with the P160 coactivator/nuclear receptor complex [3,6]. In addition, it may also play a role in the migration and differentiation of neurons [7]. In addition to nucleotide variations in the *ANKRD11* gene, KBGS is associated with 16q24.3 microdeletion or intragenic microduplication containing the *ANKRD11* gene [8]. The pathogenic mechanism in *ANKRD11* variant patients and 16q24.3 microdeletion patients may be different. Due to the loss of other genes, the phenotype in microdeletion patients shows differences.

In this study, we describe 13 cases from China of KBGS caused by variations in the *ANKRD11* gene, update the phenotype and variation spectrum of KBGS, review and analyse the previously reported literature, summarize the clinical characteristics and genetic diagnosis of KBGS, and analyse the relationship between genotype and phenotype to guide clinical diagnosis and genetic counselling.

## 2. Materials and Methods

### 2.1. Participants

We retrospectively reviewed the records of 13 Chinese KBGS patients carrying *ANKRD11* variations with no family correlation from the Department of Endocrinology, Genetics and Metabolism of Beijing Children’s Hospital, the Department of Endocrinology of Shenzhen Children’s Hospital, and the Pediatric Department of The Second Affiliated Hospital of Guangxi Medical University. All the patients met the latest recommended clinical diagnostic criteria, met two or more major criteria, or met one major standard plus two or more minor criteria [9]. The major criteria included (1) macrodontia of the upper central incisors, (2) developmental retardation related to behavioural problems or mild/moderate mental disability or learning difficulties, (3) craniofacial anomalies, (4) postnatal dwarfism, and (5) first-degree relatives with KBGS. The minor criteria included (1) conductive hearing loss caused by recurrent otitis media; (2) palatal abnormalities; (3) hair abnormalities; (4) delayed bone age (below average >2 SD); (5) skeletal anomalies, namely large anterior fontanel with delayed closure, hand abnormalities, thoracolumbar deformities, and scoliosis; (6) abnormal electroencephalogram (EEG) with or without epilepsy; (7) feeding difficulties; and (8) male cryptorchidism. This study was approved by the Ethics Committees of the Beijing Children’s Hospital of Capital Medical University. Informed consent for the clinical information and photographs of the 13 patients were obtained from the parents of the patients.

### 2.2. Genetic Test Methods

All 13 patients had anticoagulated venous blood collected for whole-exome sequencing (WES), and Sanger sequencing was performed to verify the candidate gene variations. Five families underwent WES to verify the source of variation, six patients were subjected to parental locus verification, and the source of variation was not verified in two patients. The genome reference sequence is *ANKRD11* (NG_032003.2), and the variations follow the Human Genome Variation Society (HGVS) naming guidelines (http://www.HGVS.org/varnomen, accessed on 1 November 2021). The frequency and functional annotations of the identified variations were searched in public databases and authorized databases (1000 Genome Project, human ExAC integrated, ClinVar, and HGMD professional databases). The candidate variations were evaluated and classified according to the standard guidelines of the American College of Medical Genetics and Genomics (ACMG). The bioinformatic analysis software Provean (http://provean.jcvi.org/, accessed on 14 November 2021) [10], polyPhen2 (http://genetics.bwh.harvard.edu/pph2/, accessed on 14 November 2021) [11], and Mutation Taster (http://www.mutationtaster.org/, accessed on 14 November 2021) [12] were used to predict the pathogenicity of missense variation.

### 2.3. Literature Review

The keywords “KBG”, “ANKRD11”, and “16q24.3 microdeletion/microdeletion” were used to search the PubMed, Ovid Medline, Springer, CBMD, Wanfang, and China National Knowledge Infrastructure online literature databases to screen the literature reported on KBGS patients from 1975 to May 2021. The genotypes and clinical phenotypes of KBGS patients who only had *ANKRD11* gene variations were collected.

### 2.4. Statistical Analysis

The data were analysed with the statistical analysis software SPSS 23.0. Fisher’s exact probability method was used for comparisons between groups, and the difference was statistically significant at *p* < 0.05.

## 3. Results

### 3.1. Clinical Data of the 13 Patients in This Cohort

The genotypes and clinical phenotypes of the 13 patients are shown in Table 1. The 13 patients in this study were aged from 7 months to 15 years and 3 months and included seven males and six females. In our cohort, all the patients had different degrees of cranial facial features (Appendix A). Phenotypic photos of the patients who granted informed consent are shown in Figure 1.

### 3.2. Genetic Analysis of the 13 Patients in This Study

All the WES results for the 13 patients showed heterozygous variations in the *ANKRD11* gene, including seven frameshift variations, three nonsense variations, and three missense variations. There were 11 variation sites, of which eight were not reported before: c.3562C > T (p.R1188*), c.4911delT (p.P1638Lfs*48), c.5659C > T (p.Q1887*), c.2262dupA (p.E755Rfs*27), c.5519C > T (p.A1840V), c.6122T > G (p.V2041G), c.7832A > T (p.H2611L), and c.6528_6538del(p.G2177Hfs*5). Five patients had three previously reported variation sites: c.2398_2401del (p.E800Nfs*62), c.1801C > T (p.R601*), and c.1903_1907del (p.K635Qfs*26). The source of variation in nine patients was de novo, that in two patients was from their parents, and that in two patients from Shenzhen was from unknown sources due to unavailability of parents’ blood samples. Based on the ACMG system, c.5659C > T (p.Q1887*) and c.7832A > T (p.H2611L) were rated likely pathogenic (LP), c.5519C > T (p.A1840V) and c.6122T > G (p.V2041G) were rated as variants of uncertain significance (VUS), and all the other variants were pathogenic (P).

### 3.3. Clinical Characteristics of Reported KBGS Patients

The flow chart of the selected patients is shown in Figure 2A. Only the KBGS patients with *ANKRD11* gene variation were included (*n* = 253) [2,5,8,13,14,15,16,17,18,19,20,21,22,23,24,25,26,27,28,29,30,31,32,33,34,35,36,37,38,39,40,41,42,43,44,45,46,47,48,49,50,51,52,53,54,55]. The types of variations included frameshift variation (*n* = 158), nonsense variation (*n* = 61), copy number variation (*n* = 13), missense variation (*n* = 13), splice site variation (*n* = 6), and deletion variation (*n* = 2). The source of the variation in 125 patients was de novo, that in 45 patients was from their parents, and that in 83 patients was not described. A comparison of the genotypes and clinical phenotypes of the 13 patients in our cohort and 240 previously reported KBGS patients indicated that there was no significant difference between the two cohorts, and there was no significant racial difference. No patient in our cohort had renal system anomalies, and only 14 of the previously reported patients had renal anomalies, which was not statistically significant due to the small number of samples. All 253 patients showed varying degrees of the craniofacial phenotype. Symptom or sign occurrence rates of dental anomalies (including macrodontia of the upper central incisors and other tooth anomalies), global developmental delays (especially language and walking), intellectual disability/learning difficulties (mild to moderate), limb abnormalities (especially clinodactyly of the 5th finger), and behavioural abnormalities (attention deficit hyperactivity disorder or Autistic Spectrum Disorder) were >70%. An occurrence rate >50% was associated with short stature, delayed bone age, and spinal vertebral body abnormalities. There were fewer common phenotypes, such as epilepsy, abnormal EEG, CHD, ocular anomalies, cryptorchidism, and hearing abnormalities. The frequency of clinical phenotypes is shown in Figure 2B.

### 3.4. Summary of the Phenotypic Characteristics of the Patients with Missense Variation

Among the 253 patients with gene results, 13 (~5.1%) patients—nine males and four females—had a total of 11 missense variant sites. These variations were de novo in four patients, inherited from their parents who did not meet the clinical diagnostic criteria for KBGS in six patients, and undescribed in three patients. The genotype–phenotype of these 13 patients is shown in Appendix A. The minor allele frequencies (MAFs) of these 13 missense variation sites were all <0.01. The position and conservation analysis of the 11 missense variation sites are shown in Figure 3.

Fisher’s exact probability method was used to analyse the difference in the frequency of clinical phenotypes between 218 cases of *ANKRD11* truncated variation (including nonsense variation and frameshift variation) and 13 cases of missense variation. The results showed that the differences between the two groups of patients in global developmental delays and intellectual disability/learning difficulties were statistically significant (*p* < 0.05). The statistical table is shown in Appendix A.

## 4. Discussion

### 4.1. Prevalence of KBGS

Currently, worldwide, the prevalence of KBGS is still unclear; although only 240 cases of KBGS caused by *ANKRD11* gene variation have been reported, some studies have shown that *ANKRD11* variation accounts for approximately 1% of patients with unclear aetiology of growth retardation [56]. The number of patients with KBGS reported at present may be far lower than those actually carrying *ANKRD11* gene variation, which may be due to the incomplete understanding of the disease by many doctors, the variability in disease phenotypes, and the mild phenotypes of some patients. In the past ten years, WES has greatly promoted the molecular diagnostic rate of patients with growth retardation and mental retardation, but most studies are single-centre studies or small cohort reports.

### 4.2. Clinical Malformation of KBGS

Several clinical diagnostic criteria for KBGS have been proposed, but there is still no international consensus regarding KBGS. In the 253 KBGS patients, the clinical phenotypes were mainly concentrated in craniofacial abnormalities, growth and developmental abnormalities, skeletal system abnormalities, and nervous system abnormalities. Since some symptoms can only be observed at a later stage, this results in delayed diagnosis. In infancy and childhood, patients often show growth retardation, feeding difficulties, delayed closure of anterior fontanelle. If these symptoms are accompanied by craniofacial anomalies, limb anomalies, or anomalies of the above related systems, we should be alert to KBG syndrome. At present, there are few reports on elderly patients with KBGS, and the long-term prognosis of patients remains to be observed. The oldest patient, aged 46 years, showed moderate hunchback (kyphosis) and osteopenia [2]. Yoda mice with a homozygous missense variation in *ANKRD11* showed kyphosis and decreased bone mineral density with age, and the characterization is related to sex: 80% of the elderly female Yoda mice showed kyphosis [4].

### 4.3. Relationship between Genotype and Phenotype

This study reported three patients with missense variation (P10, P11, P12). Patient 11 and Patient 12 carry *ANKRD11* missense variation localized in the important C-terminal domain of ANKRD11, which may affect the degradation and abundance of the ANKRD11 protein. Patient 10 carries *ANKRD11* missense variation c.5519C > T (p.A1840V). The patient’s clinical phenotype was consistent with the existing clinical diagnostic criteria, and his mother carried the same variation and only showed the phenotype of a short fifth finger. The pathogenicity of a missense variation in the *ANKRD11* gene needs to be considered cautiously; clinical phenotypes caused by other genes need to be excluded, and functional verification may be necessary. Zhang et al. [50] found that a missense variation, c. 6427C > G (p.L2143V), in the *ANKRD11* gene did not lead to abnormal expression of ANKRD11 protein and RNA, but could not restore the regulatory effect of *P21*, which is an important factor in regulating cartilage differentiation. At present, only 13 cases of KBGS caused by missense variation of the *ANKRD11* gene have been reported. Comparing the clinical phenotypes of 218 patients with truncated variation and 13 patients with missense variation in the *ANKRD11* gene, the frequency of global developmental delays and intellectual disability/learning difficulties in patients with truncation variation was higher than that in patients with missense variation in the *ANKRD11* gene, which may be related to the loss of function of the ANKRD11 protein in nervous system development caused by truncation variation. Based on the small number of patients with missense variation, larger samples are needed.

### 4.4. Clinical Therapy of KBGS

At present, treatment of KBGS is mainly symptomatic, involving comprehensive clinical multidisciplinary therapy. Several reports have confirmed the effectiveness of growth hormone therapy for short stature [43,49]. In our cohort, P10 was treated with growth hormone for 1.3 years, and his height standard deviation was reduced from −3.2 SD to −1.7 SD. However, there is no specific treatment for developmental disorders. KBGS mainly affects the cranial nervous system. As a chromatin regulator, the *ANKRD11* gene plays an important role in the development of the cranial nervous system. Recent studies have found that homozygous deletion of *ANKRD11* in the neural crest is embryonic lethal, while heterozygotes exhibit a craniofacial phenotype similar to KBGS patients [57]. This suggests that the *ANKRD11* gene plays an important role in the neural crest. A group of diseases caused by abnormal neural crest development are collectively referred to as neurocristopathies (NCPs), which can affect multiple systems (craniofacial, heart, limbs, etc.) [58]. However, how *ANKRD11* plays a role in the neural crest remains unclear. In addition, some researchers suggest that *ANKRD11* gene variation should be classified as a neurodevelopmental disorder and belong to the category of chromatin disease because the phenotypes of KBGS, CdLS, and Coffin–Siris syndrome (CSS) overlap greatly [52]. CdLS is caused by genetic variants that affect subunits or regulators of the cohesion complex [59]. Although some studies have listed the *ANKRD11* gene as the causative gene of CdLS, the relationship between the *ANKRD11* gene and the cohesion complex is still unclear. Exploring the molecular mechanism of the *ANKRD11* gene will help in studying treatment options.

## 5. Conclusions

In summary, this study reported the genotypic and phenotypic characteristics of patients with KBGS from the largest sample in China, enriched the variation spectrum of the *ANKRD11* gene, and provided a basis for genetic counselling and clinical diagnosis of patients with KBGS. We reviewed and analysed the phenotypes of patients with KBGS reported in the literature. The clinical phenotypes were mainly concentrated in craniofacial abnormalities, growth and developmental abnormalities, skeletal system abnormalities, and nervous system abnormalities. We observed that patients with truncation variation in the *ANKRD11* gene had a higher frequency of global developmental delays and intellectual disability/learning difficulties than those with missense variation. However, further studies are needed in larger groups to clarify the genotype–phenotype relationship. In addition, the molecular mechanism of the *ANKRD11* gene as a chromatin regulator needs to be explored. Better treatment options and improvement of quality of life for patients with KBGS are also problems that patients and their families expect to be solved.

## Figures and Tables

**Figure 1 jpm-12-00407-f001:**
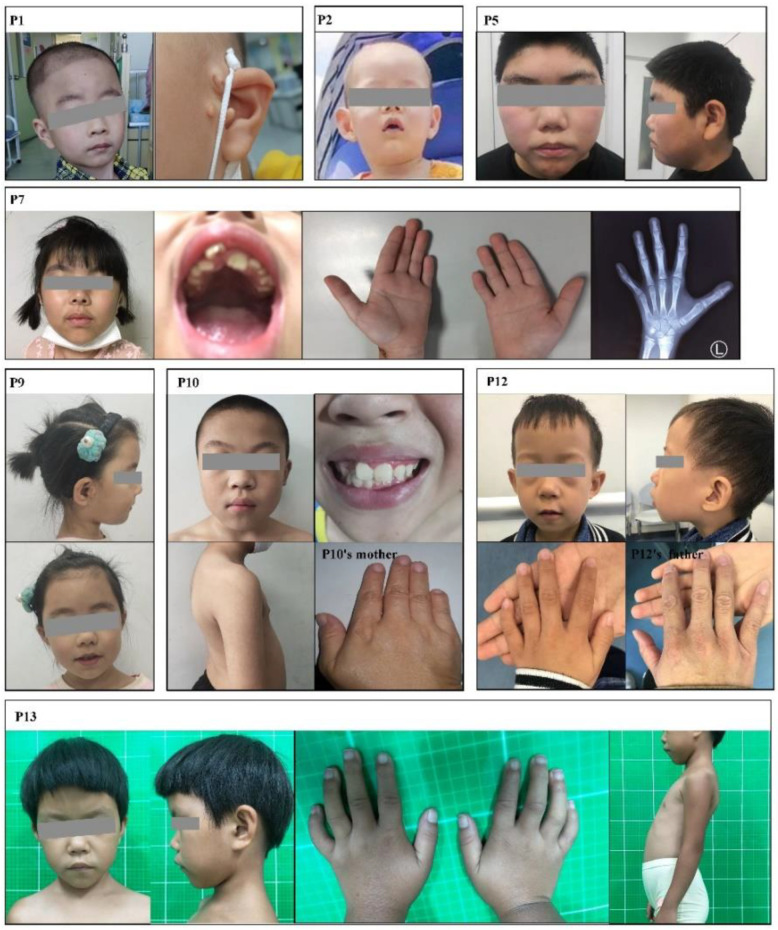
Clinical characteristics of patients in this study.

**Figure 2 jpm-12-00407-f002:**
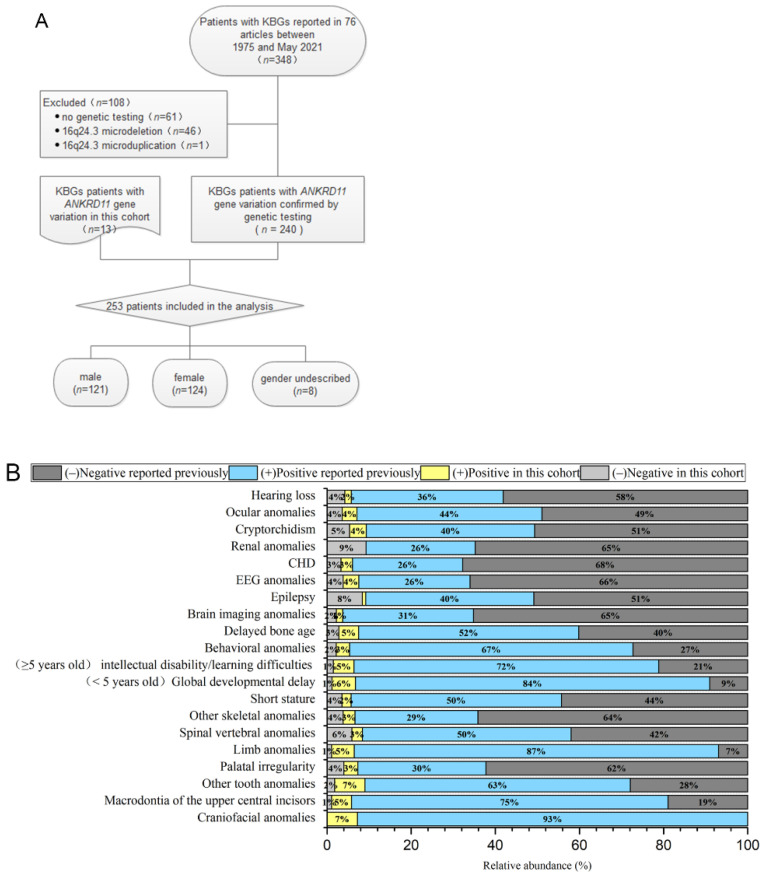
(**A**) The process of screening the patients. (**B**) Clinical phenotype frequency of the 253 KBGS patients with *ANKRD11* gene variation only.

**Figure 3 jpm-12-00407-f003:**
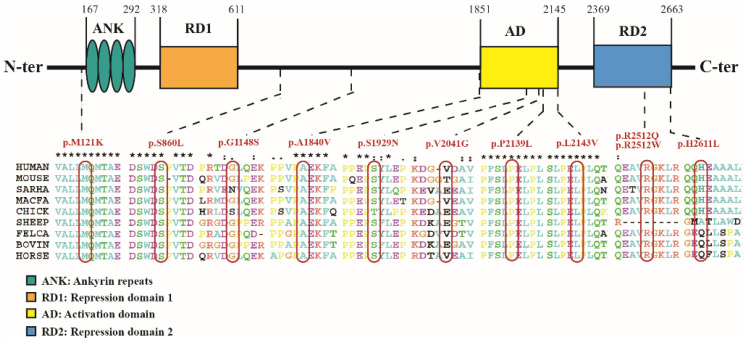
Position and conservation analysis of the 11 missense variation sites in *ANKRD11* gene.

**Table 1 jpm-12-00407-t001:** Clinical and genetic analysis of 13 cases in this cohort.

Patient	P1	P2	P3	P4	P5	P6	P7	P8	P9	P10	P11	P12	P13
**Variant**	c.3562C > T (p.R1188*)	c.2398_2401del (p.E800Nfs*62)	c.4911delT (p.P1638Lfs*48)	c.5659C > T (p.Q1887*)	c.1801C > T (p.R601*)	c.1903_1907del (p.K635Qfs*26)	c.1903_1907del (p.K635Qfs*26)	c.1903_1907del (p.K635Qfs*26)	c.2262dupA (p.E755Rfs*27)	c.5519C > T (p.A1840V)	c.7832A > T (p.H2611L)	c.6122T > G (p.V2041G)	c.6528_6538del (p.G2177Hfs*5)
**Variation source**	De novo	De novo	De novo	Na	De novo	De novo	Na	De novo	De novo	Mother	De novo	Father	De novo
**Age at diagnosis**	3y4m	1y2m	5y10m	3y8m	15y3m	7m17d	7y8m	3y6m	6y4m	12y	3y6m	4y8m	7y
**Gender**	Male	Male	Male	Female	Male	Male	Female	Female	Female	Male	Female	Male	Female
**ACMG**	P	P	P	LP	P	P	P	P	P	VUS	LP	VUS	P
**Birth history**	+SGA	−	−	−	−	Na	−	−	−	−	−	+SGA	−
**Perinatal issues**	−	+Feeding difficulties	+Feeding difficulties, vomit	+Feeding difficulties, vomit	+Feeding difficulties	Na	−	+Hypotonia	+Feeding difficulties	−	Na	+Feeding difficulties, large fontanelles	−
**Craniofacial anomalies**	+	+	+	+	+	+	+	+	+	+	+	+	+
**Dental anomalies**													
Macrodontia	+	+	+	+	+	Na	+	+	−	+	+	+	Na
Other dental anomalies ^a^	+	−	+	+	−	Na	+	+	+	−	Na	+	Na
**Skeletal anomalies**													
High palate	+	Na	−	+	−	+	−	−	+	−	+	+	+
Clinodactyly of the 5th finger	+	+	−	+	−	Na	+	+	+	−	+	+	+
Short finger/small hand	−	−	−	−	+	Na	+	+	+	−	−	−	−
scoliosis/kyphosis	−	−	−	Na	Na	Na	Na	−	−	+	−	−	+
Other skeletal anomalies ^b^	+	−	+	Na	Na	Na	Na	+	–	Na	–	–	+
**Short stature**	−, 92.2 cm (−1.8 SD)	−, 77 cm (−0.6 SD)	+, 99.8 cm (−3.8 SD)	+, 91 cm (−2.7 SD)	−, 160 cm (−1.6 SD)	−, 69.5 cm (−0.2 SD)	−, 123 cm (−0.7 SD)	−, 99 cm (−0.6 SD)	−, 114 cm (−1.5 SD)	+, 132 cm (−3.2 SD)	+, 92 cm (−2 SD)	+, 101 cm (−2 SD)	−, 116.5 cm (−1.9 SD)
**Growth retardation**													
Global development delay	+	+	+	+	−	+	+	+	+	−	Na	+	+
Speech and language development delay	+22 mo	+	−	−	−	Na	+4 yr	−	Na	+22 mo	+	−	+
Intellectual disability/learning difficulties	+	+	+	−	+	+	+	+IQ79	−	+	+IQ74	−	+
Behavioural anomalies	−	−	−	+Hyperactivity	Na	Na	+Hyperactivity, short attention span	−	+Short attention span	+Hyperactivity, short attention span	+less communication	+short attention span, timid	Na
Delayed bone age	Na	Na	+	+	+	Na	−Advanced	−	+	+	−	Na	Na
**Neurological abnormalities**													
Epilepsy	−	−	−	−	Na	−	−	+	−	−	Na	−	−
EEG anomalies	Na	+	Na	Na	Na	−	Na	+	Na	Na	Na	−	Na
Brain imaging anomalies	Na, pituitary MRI(−)	+Left ventricle slightly wider	−	Na	+Small pituitary	Na	Na, pituitary MRI(−)	-	Na	Na, pituitary MRI(−)	Na	−	Na
**Ocular anomalies**	−	Na	−	−	+Visual field defect	Na	+Strabismus	Na	+Strabismus	−	−	+Astigmatism	+Myopia
**Hearing loss**	−	Na	−	−	+	−	−	−	+	−	Na	+Recurrent otitis media	−
**CHD**	+PDA	−	+VSD	Na	−	+VSD	+VSD	−	Na	−	−	+	−
**Cryptorchidism**	+	−	+	/	−	−	/	/	/	+	/	−	/
**Renal anomalies**	−	−	−	Na	−	Na	Na	Na	Na	Na	Na	−	Na
**1st degree relative with KBG**	−	−	−	Na	−	−	−	−	−	−	−	−	−
**Additional features**	Deviated nasal septum, sinusitis, preauricular skin tag, supra-auricular pit	Head pilomatrixoma, large fontanelles	Inguinal hernia, micropenis	−	Delayed puberty, micropenis, small testicles, hypogonadotropic hypogonadism, enlarged vestibular aqueduct, obesity	−	Central precocious puberty	Supra-auricular pit	−	–	−	−	−

+, Positive Phenotype; −, Negative Phenotype; Na, Data non-available; SGA, small for gestational age infant; PDA, Patent ductus arteriosus; VSD, Ventricular septal defect; ACMG, the American College of Medical Genetics and Genomics; VUS, Uncertain significance; P, Pathogenic; LP, Likely pathogenic. a, Other dental anomalies include misalignment of teeth (P1,P3,P4,P7,P8,P9,P10), oligodontia (P9), hypoplasia of dental enamel (P12). b, Other skeletal anomalies include pectus excavatum (P1), spina bifida occulta (P3), caudal appendage (P8), joint stiffness and shallow acetabular fossae (P13). Bold: represents large class.

## Data Availability

The data presented in this study are available on request from the corresponding author.

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
