# Peer review of "Genetic and Phenotypic Spectrum of KBG Syndrome: A Report of 13 New Chinese Cases and a Review of the Literature"

_jpm, 2022, doi:10.3390/jpm12030407_

Round 1
Reviewer 1 Report
My comments and suggestions are:
line:
79-83 "...13 Chineese KBGS patients carrying ANKRD11 variations" if any of these patients has earlier a suspition of KBG syndrome?
83-85 Retrospectively the patients met clinical criteria of KBGS but if KBGS were not suspected earlier so what were the clinical signs qualifying those patients to WES?
179-182 "Among 253....... a total of 11 missense variant sites......de novo in 4 patients, inherited from their parents who did not meet the clinical diagnosis criteria for KBGS in 6 patients...." And what about truncated variations, inherited from parents, if the parents with these variations had the clinical signs of KGBS? is there any difference in group of parents with these two variations?
222-225and what about patient's 12 father? if he showed any signs of KGBS?
233-238
you compare the frequency of intellectual disability/learning difficulties in group of 240 patients with truncated variation in the ANKRD11 gene to the grup of 13 patients with missense variations and make conclusions but in the group of 13 patients with missense variations are included 2 parents -one without intellectual disability and the second with borderline IQ what make influence on the frequency of ID in that small group. So as you wrote further studies are needed in larger group.
The very intersting in this paper is that authors listed clinical signs in patients with missense variations and then compared two group of patients with two different kind of variations - truncated and missense and found some differences in frequency of clinical signs as intellectual disability/learning difficulties.
Reviewer 2 Report
In this paper, the authors report 13 news cases of KBG syndrome. They describe the clinical phenotype and genotype of this patients. While these patients share common clinical features described previously in the literature, the authors were able to expand the phenotype. They also identified novel variants linked to KBG syndrome, including frameshift, missense and nonsense mutations. This paper is relevant and will greatly contribute to the diagnosis of patients carrying variants in ANKRD11 gene and is suitable for publication. I have only minor comments/suggestions:
- page 2, line 50 then line 63: the authors appear to contradict themselves regarding the pathophysiology of KBG syndrome. They first say variants in ANKRD11 that lead to haploinsufficiency are the cause of this syndrome; then they say that dominant-negative effect is the main pathogenic mechanism. In the current format, it is confusing to the reader as it gives the impression the literature says variants lead to haploinsufficiency then they say dominant-negative effect. As KBG syndrome is linked to nonsense, missense and frameshift mutations, both mechanisms are possible.
- In the discussion, the authors say that KBG syndrome is a neurodevelopmental disorder. However, the phenotype points to multiple skeletal defects. Likely, this is a structural skeletal birth defect and neurodevelopmental disorder. Since expression data for ANKRD11 during embryonic development is not clear (expression in the adult brain and glia may not correlate to embryonic expression), classification at this stage is not possible. Some skull birth defects (e.g. craniosynostosis) can lead to intellectual disability as the brain cannot grow because of the skeletal defect. Thus, it would be informative if the authors added some discussion about this issue in page 3, second paragraph (lines 240-253).
Reviewer 3 Report
This is a comprehensive presentation of 13 KBG syndrome patients from China. The authors have clearly presented all major clinical features of these patients. And they successful identified eight new disease-causing variants in ANKRD11 gene, and classified all the variants based on ACMG criteria. Authors have also done great job with reviewing and analyzing the phenotypes of reported patients with KBG syndrome.
Two minor comments for the authors.
Comment 1: Lines 219-221: Authors don’t provide justification for second half of following sentence “Patient 11 and Patient 12 carry ANKRD11 missense variation localized in the important C-terminal domain of ANKRD11, which may affect the degradation and abundance of the ANKRD11 protein.” Are there any previous reports degradation of ANKRD11 protein due to C-terminal mutations? If so, authors must cite that reference.
Comment 2: Line 129: Authors may want to consider changing “cranial facial features” to “craniofacial features”.
Author Response
Dear reviewer:
Thank you for your suggestion. As suggested by reviewer, we have added the suggested content to the manuscript.
Point 1: Lines 219-221: Authors don’t provide justification for second half of following sentence “Patient 11 and Patient 12 carry ANKRD11 missense variation localized in the important C-terminal domain of ANKRD11, which may affect the degradation and abundance of the ANKRD11 protein.” Are there any previous reports degradation of ANKRD11 protein due to C-terminal mutations? If so, authors must cite that reference.
Response 1: Thanks for your reminder, we have added the relevant citation.
Point 2: Line 129: Authors may want to consider changing “cranial facial features” to “craniofacial features”.
Response 2: Thanks for your suggestion, we have made changes in response to your question.
This manuscript is a resubmission of an earlier submission. The following is a list of the peer review reports and author responses from that submission.